# Adenovirus Infection in Pediatric Hematopoietic Cell Transplantation: A Challenge Still Open for Survival

**DOI:** 10.3390/jcm11164827

**Published:** 2022-08-18

**Authors:** Simone Cesaro, Fulvio Porta

**Affiliations:** 1Pediatric Hematology and Oncology, Department of Mother and Child, Azienda Ospedaliera Universitaria Integrata Verona, Piazzale Aristide Stefani, 1, 37126 Verona, Italy; 2Pediatric Oncohematology and Bone Marrow Transplant Unit, Children’s Hospital, ASST Spedali Civili of Brescia, 25123 Brescia, Italy

**Keywords:** adenovirus infection, adenovirus disease, cidofovir, preemptive therapy, risk factors, survival

## Abstract

Human Adenovirus (HAdV) infection occurs in 14–16% of patients in the early months after pediatric hematopoietic cell transplantation (HCT) and this correlates with a higher risk of developing HAdV disease and overall 6-month mortality. The main risk factors for HAdV infection are T-cell depletion of the graft by ex vivo CD34+ selection or in vivo use of alemtuzumab or anti-thymocyte serum, the development of grade III-IV graft versus host disease (GVHD), the type of donor (unrelated donor, cord blood, haploidentical, or HLA mismatched parent), and severe lymphopenia (<0.2 × 10^9^/L). The prevention of HAdV disease is based on early intervention with antivirals in the asymptomatic patient when the permitted viral load threshold in the blood (≥10^2–3^ copies/mL) and/or in the stool (10^9^ copies/g stool) is exceeded. Cidofovir, a monophosphate nucleotide analog of cytosine, is the primary drug for preemptive therapy, used at 5 mg/kg/week for 2 weeks followed by 3–5 mg/kg every 2 weeks. The alternative schedule is 1 mg/kg every other day (three times/week). Enhancing virus-specific T-cell immunity in the first months post-HCT by donor-derived or third-party-derived virus-specific T cells represents an innovative and promising way of intervention, applicable both in prevention and therapeutic settings.

## 1. Introduction

Human Adenovirus (HAdV) may cause infection in the early months after hematopoietic cell transplantation (HCT) and, together with cytomegalovirus (CMV), Epstein–Barr virus (EBV), Human Herpes virus 6 (HHV-6) and BK polyomavirus (BKPyV), is responsible for most viral complications in the first months after transplant [1,2,3,4]. The occurrence of HAdV viremia within 6 months of HCT viremia is associated with at least one other viral infection, mostly CMV, in 78% of pediatric HCTs and 87% of adult HCTs [5].

In a survey for the European Society for Blood and Marrow Transplant (EBMT) conducted in 2018 for the period 2013–2016 and involving 91 transplant centers, the annual incidence of HAdV infection (intended as any positivity of HAdV viremia) had a median of 7%, ranging from 5.8% to 9%, with a marked difference between pediatric and adult patients, 15.4% versus (vs.) 4.1% [6].

A multinational, multicenter, European study (AdVance study) conducted in 50 European HCT centers in the period 2013–2015 reported the cumulative incidence of HAdV infection in the first 6 months after HCT and using any type of positive assay for HAdV of 32% and 6% in pediatric and adult patients, respectively. The cumulative incidence of significant HAdV viremia (HAdV ≥10^3^ copies/mL) was 14% in pediatric patients and 1.5% in adult patients [7].

In a companion survey performed in the United States from 2015 to 2017 in 15 pediatric HCT centers enrolling 1230 patients and 6 adult HCT centers enrolling 1815 patients, the incidences of HAdV infection, HAdV viremia, and HAdV viremia ≥ 10^3^ copies/mL within 6 months after the first allogeneic (allo) HCT were 23%, 16%, and 9% for pediatric patients, and 5%, 3%, and 2% for adult patients, respectively. Importantly, the overall 6-month mortality rate was higher in patients who developed HAdV viremia ≥ 10^3^ copies/mL compared with those who did not, being 17% vs. 10% in pediatric patients and 23% vs. 16% in adult patients [5].

## 2. Biology

HAdV was named after its isolation from human adenoids in 1953. It is a medium-sized (70–100 nm), nonenveloped, double-stranded DNA virus belonging to the Adenoviridae family. The HAdV DNA genome is contained in an icosahedral shell composed of 240 hexon capsomeres, 12 pentameric capsomers, and 12 fibers. Figure 1 schematically represents the structure of HAdV. HAdVs are classified into seven species, from A to G, according to the content of guanine and cytosine in their DNA, while types are defined serologically up to AdV 51 and based on genetic characteristics for the other types [8]. Table 1 shows the classification of the known genotypes, grouped according to the species [9]. Species D is the largest one and includes more than 70 types, followed by species B, with 16 types. About one-third of HAdV types are associated with human disease and show different tropism for tissues and organs.

Primary infection occurs during infancy and usually can be asymptomatic or simulate a self-limiting flu-like infection or intestinal gastroenteritis, while other severe manifestations, such as myocarditis, encephalitis, and meningitis, are rare [10].

The transmission of the virus occurs after contact with an infected individual through respiratory droplets, direct contact of eye mucosae with infected fluids, or by the fecal–oral route. HAdVs are resistant to gastric and biliary secretions and also to dry environments for several weeks; moreover, they are resistant to many disinfectants, being nonenveloped viruses. The decontamination of surfaces requires alcohol solutions (85 to 95%) for at least 2 min or sodium hypochlorite for 10 min [11].

In immunocompetent children, the susceptibility to HAdV infection starts after 6 months of age, because in the first months of life the protection is conferred by maternal antibodies, and peaks by 5 years of age, concurrently with the progressive exposure to life and activities in the community. In this period, it is estimated that 7–8% of respiratory infections are caused by HAdV. After infanthood, most adults are immunized as revealed by the presence of neutralizing antibodies for HAdV in healthy blood donors or commercially available preparations of polyclonal intravenous immunoglobulins [12]. Once HAdV enters T lymphocytes or other cells of several tissues or organs (tonsil, adenoids, gut mucosa, lung, and brain) through receptors such as the CD46 receptor or integrins, it persists throughout life and the reduction of specific T-cell immunity may trigger the reactivation of viral replication and secondary infection and HAdV disease. Unlike other DNA viruses, HAdV genome DNA is not integrated into the host cell DNA but remains in an episomal state. Episomes are non-integrated extrachromosomal closed circular DNA molecules that may be replicated in the nucleus.

## 3. Diagnosis

The diagnosis of HAdV infection is based on the assessment of the virus in different biological samples such as peripheral blood, respiratory fluids or secretions, nasopharyngeal swab, stool, and tissue biopsy. Unlike the immunocompetent host, the use of serology is not useful in this setting [10]. The PCR-based method has largely replaced the conventional methods of detection of viral antigens or of viral culture that have a lower sensitivity, and for viral culture, a longer time to readout [13]. Due to its sensitivity and specificity, qualitative and quantitative PCR assays for HAdV DNA are considered the reference diagnostic methods due to the quick time to get the result and the efficacy in detecting all HAdV types [14,15,16]. The use of primers for the conserved parts of the HAdV genome, such as the hexon capsid protein gene and the fiber protein gene, periodically updated based on new HAdV types identified by genomic analysis, permits the detection of all HAdV types reliably and maintains an adequate sensitivity [4]. Importantly, in the case of screening patients at higher risk of progression to HAdV disease with a quantitative PCR method in peripheral blood, the lower limit of detection should be in the range of 10^2^ copies/mL to allow preemptive treatment while the patient is asymptomatic [17]. The identification of HAdV at the level of serotype or strain, while it is important for epidemiological investigations or the documentation of nosocomial outbreaks, is not usually performed in daily practice because it is quite irrelevant in the choice of treatment. The only exception is to search for an infection caused by HAdV C that may help introduce ribavirin [18,19].

## 4. Clinical Symptoms and Definitions

HAdV disease usually occurs between 2 and 3 months post-HCT, with the most frequent symptoms being fever, enteritis, elevated liver enzymes, and secondary pancytopenia. In the absence of effective T-cell immunity, HAdV replicates unchallenged, involving multiple sites or organs, determining from mild gastroenteric or respiratory symptoms to severe manifestations, including hemorrhagic enteritis or cystitis, pneumonia, hepatitis, nephritis, encephalitis, myocarditis, and up to multiorgan failure and death. In patients with HAdV disseminated disease, the fatality rate can be as high as 60–80%, and some cases of fulminant evolution in a few days after diagnosis have been reported [16]. This underlines the importance of an early diagnosis of impending HAdV disease and the need to intervene to block the viral replication. HAdV disease is defined as a local or systemic (invasive) disease, depending on whether the replication of the virus is limited to a single organ/tissue or involves the blood. Moreover, HAdV disease is defined as probable or proven depending on the presence of 2 or 3 of the following criteria, respectively: (a) documented HAdV infection in blood or tissue (viral replication determined by DNAemia, antigen detection, or viral culture), (b) clinical signs and symptoms consistent with HAdV infection, (c) histological documentation of HAdV tissue or cell involvement. All three criteria must be met for proven HAdV disease. Disseminated HAdV disease is defined as multiple-organ involvement (pneumonia, hepatitis, encephalitis, and retinitis) in the presence of two or more HAdV-positive PCR assays in peripheral blood and other sites tested (cerebrospinal fluid, bronchoalveolar lavage fluid, respiratory secretions, or urine), and in the absence of alternative identifiable causes of disease [11,14]. Death in the context of multi-organ involvement with persistently high or increasing adenoviral load is attributed to HAdV disease.

## 5. Risk Factors

Classical risk factors for HAdV infection and disease in allo-HCT are the pediatric age and the procedures or treatments inducing severe compromission of T cell immunity and or severe lymphopenia in the early phase post-HCT. These factors are T-cell depletion of the graft by ex vivo CD34+ selection or in vivo use of alemtuzumab or anti-thymocyte serum, the development of grade III-IV graft versus host disease (GVHD), the type of donor (unrelated donor, cord blood, haploidentical or HLA mismatched parent), and severe lymphopenia in the first months after allo-HCT(<0.2 × 10^9^/L) [7,9,14,20,21]. In a retrospective North American study among 624 adult and pediatric patients, the incidence of ADV infection was 8% in the T-cell depleted HCT versus (vs.) 4% in the unmanipulated conventional HCT; moreover, among the T-cell depleted recipients, the incidence of HAdV infection was 15% in children vs. 5% in adults. Young age and acute GVHD were risk factors for HAdV infection, while HAdV infection, defined by at least two consecutive PCR results ≥ 10^3^ copies/mL blood, was a risk factor for mortality. In the group of T-cell depleted patients, HAdV disease occurred in 3.5% of patients vs. 0.4% of patients who did not receive a T-depleted HCT, with an overall attributable mortality of 27% [22]. In a retrospective European study enrolling 1736 pediatric and 2540 adult patients, the incidence of HAdV infection, defined by a viral load ≥ 10^3^ copies/mL blood, was 14% in pediatric patients and 1.5% in adult patients. In this study, the risk factors for HAdV infection were younger age, T-cell depletion, and donor type (unrelated donor or mismatched related donor) [7].

More recently, the weekly surveillance of HAdV viral load in the stool in the first 100 days post-HCT showed that a concentration ≥ of 10^6^ copies/g of faeces is predictive of HAdV infection and disseminated disease and may anticipate HAdV viremia in a median of 11 days [4,23,24,25]. The risk of developing disseminated HAdV disease is even higher for patients who have the shedding of HAdV in their stool before starting the conditioning regimen for HCT. In a prospective study on 304 pediatric HCT patients, 42 patients (14% of the cohort) presented HAdV shedding in the stool before HCT; in this group, HAdV viremia was higher than that observed in the cohort without viral shedding in the stool, 33% vs. 7% [26]. These findings confirm that the gastrointestinal tract is a reservoir for the persistence and replication of HAdV in children that rapidly expands before spreading to the blood and organs [27]. Unlike in children, the role of the gastrointestinal tract as a reservoir for persistence, replication, and expansion of HAdV is not certain in adult immunocompromised patients, and screening the stool in these patients is not recommended.

Uncontrolled HAdV replication is not only a risk factor for progression to HAdV disease but also a risk factor for overall survival. In a prospective study among 238 pediatric allo-HCT, HAdV viral load ≥10^4^ copies/mL resulted in an independent risk factor for poor survival [28].

A retrospective analysis of the factors predicting the outcome in 241 pediatric HCT patients with HAdV infection found that the higher the peak level and longer duration of HAdV viremia correlated with 6-month mortality independently from other known risk factors such as lymphopenia. Higher values of the area-under-viremia-curve over 16 weeks (AUC 0–16), a surrogate indicator built with the weekly results of HAdV viremia, were independent risk factors for all causes of mortality. Therefore, lowering the viral load and shortening the duration of viral replication has the potential to reduce mortality in a population at risk of HAdV [29]. Table 2 shows the main risk factors for HAdV infection and disease.

## 6. Prevention

The risk of HAdV infection is strictly related to the severe immunosuppression performed for the transplant procedure, and this risk remains active until the start of T-cell immunity recovery. The window of maximum risk is the second–third month after the transplant. Considering that the HAdV infection is the consequence of the host-virus reactivation and does not originate from external infected contacts, the classical measures, such as isolation, hand hygiene, and room disinfection, are not sufficient to prevent HAdV infection. Several studies showed that during this period, it is possible to select the population at higher risk of progression to HAdV disease by monitoring the viral replication in the blood and the stool by a quantitative PCR method. The gut is a reservoir of HAdV during infancy, and therefore the quantification of HAdV in the stool is more suitable for pediatric HCT patients. The quantification of HAdV in the blood is possible both in adults and in pediatric HCT, but the screening for HAdV replication is more often used in pediatric centers given the higher incidence of infection and disease [6,30]. The aim of the weekly monitoring of HAdV replication in the early phase post-HCT is to start antiviral medical therapy that can stop or decrease viral replication. A scheme of this approach is shown in Figure 2.

## 7. Treatment

HAdV infection represents a life-threatening complication in patients who undergo allo-HCT. In these patients, it is crucial to identify early HAdV infection and adopt measures to reduce or stop viral replication, preventing it from becoming disseminated. As in other viral infections in the HCT setting, the possible approaches are prophylaxis, preemptive treatment, and therapy of established HAdV disease. The therapeutic options to counteract HAdV replication and HAdV disease are the tapering of post-transplant immunosuppression, the use of antivirals, and the enhancement of immune recovery by adoptive immunotherapy. The tapering of immunosuppression has a limited role in the management of HAdV infection because it is possible only in HCT performed with an unmanipulated graft where post-transplant immunosuppression is required, while the higher incidence of this complication is observed in the T-cell depleted HCT that usually does not include immunosuppressive drugs after transplant. Whenever possible, the tapering of immunosuppression is recommended in a lymphopenic patient with HAdV viremia ≥ 10^3^ copies/mL and a CD3+ count < 0.2–0.3 × 10^9^/L or with a virus concentration in the stool above the critical threshold of 10^6^ copies/g [14,31]. The use of antivirals is indicated for preemptive therapy and the treatment of disease. Preemptive therapy is the treatment of the asymptomatic patient to block or reduce viral replication and prevent the progression to overt disease until specific immune reconstitution against HAdV is achieved. In this setting, the most indicated drug is cidofovir, a monophosphate nucleotide analogue molecule of cytosine that is phosphorylated intracellularly to diphosphate and has a higher affinity for viral DNA polymerase compared with cellular DNA polymerases. Diphosphates of cidofovir compete with cellular nucleoside triphosphates and inhibit viral DNA replication by their competitive incorporation into DNA strands [32]. Importantly, cidofovir is effective against all HAdV species, although some resistant mutants have been reported in vitro studies [33]. The antiviral efficacy of cidofovir as a preemptive intervention against HAdV has been observed in several studies, while its efficacy is lower when used for the treatment of established disease [11,34,35]. Cidofovir is the primary drug for preemptive therapy and it is used at 5 mg/kg/week for 2 weeks followed by 3–5 mg/kg every 2 weeks. In cases where probenecid is not available or not tolerated, the alternative schedule is 1 mg/kg every other day, with or without probenecid (three times/week) [17]. The main route of excretion is the renal one, and the drug is found unchanged in the urine of more than 90% of patients. The route of excretion explains the intrinsic nephrotoxicity of cidofovir. The drug penetrates from the blood into kidney cells by organic anion transporters present on the basolateral membrane of the renal tubular cells and, due to a lower efflux into the tubule lumen, may achieve a high, toxic concentration, inducing cell death. The prevention of nephrotoxicity is based on hyperhydration that dilutes the intracellular concentration of cidofovir and on probenecid, an organic acid molecule that competes with cidofovir for the kidney’s cell organic anion transporter, reducing the intracellular accumulation [36,37]. The treatment is continued until virological response and adequate immune reconstitution are achieved. The recommended threshold is: HAdV viremia < 400 copies/mL, CD3+ count > 0.3 × 10^9^/L [14,17,31]. The limitations of cidofovir are the modest bioavailability that may prevent achieving an effective concentration of the active phosphorylated metabolites inside the infected cells and the absence of an oral formulation with the consequent need for intravenous administration. The efficacy of preemptive treatment with cidofovir in the allo-HCT setting is variable, but still today, it represents the only available option to apply early drug intervention and it has been proposed in several studies and recommended in guidelines [13,14,17,28,31]. The effectiveness of cidofovir cannot be separated by host T-cell reconstitution and acquisition of the specific HAdV cell immunity that usually requires weeks to months after HCT. Therefore, the preemptive use of cidofovir aims to stabilize and control the viral replication in the early weeks post-HCT for the time needed for immune recovery. Late introduction of cidofovir, once HAdV viremia is ≥10^4^ copies/mL or HAdV disease is established, was initially associated with a response or survival rate of up to 80% but these results have not been confirmed in other series where more limited efficacy and higher mortality have been reported [13,34,38]. The development of brincidofovir, a lipid conjugate of cidofovir, orally available, that is converted intracellularly into the active antiviral cidofovir diphosphate, has raised many expectations [39]. This lipid conjugation results in good intestinal absorption, higher intracellular concentrations of active drug, reduced uptake by renal tubular cells, and increased antiviral potency against several double-stranded DNA viruses such as cytomegalovirus, HAdV, BK polyomavirus, vaccinia virus, and *Molluscum contagiosum* virus. Several retrospective or prospective phase II studies documented the efficacy of brincidofovir in reducing HAdV replication irrespective of lymphocyte count, patient immune recovery status, and previous treatments with other antivirals, including cidofovir, both in adult and pediatric patients [40,41,42]. Interestingly, brincidofovir has no significant nephrotoxicity or myelotoxicity while the most common adverse effect is intestinal toxicity (diarrhea) that, in HCT patients, is not easily distinguishable from intestinal GVHD. The subsequent phases of development of the drug as a prophylaxis for CMV and preemptive therapy for HAdV failed to achieve the primary endpoints [43], so the drug is not available anymore, also for compassionate use. Another drug tested for HAdV infection is ribavirin. Ribavirin is a nucleoside analogue of guanosine that has in vitro activity against DNA and RNA viruses by inhibiting viral polymerase activity, viral RNA capping, and increasing the mutation rate in newly synthesized DNA. In vitro analysis shows that ribavirin is only effective against the HAdV C type, but its efficacy in vivo is debated or, at least, not well-proven [11]. Despite that, considering the low nephrotoxicity, the use of ribavirin may be added to cidofovir in the case of refractory or slow-responding infection by HAdV type C [31]. Other antivirals, such as ganciclovir or foscarnet, are not used against HAdV infections because they have a questionable effect or no effect at all.

Figure 2 summarizes the monitoring and the early treatment of allo-HCT patients at high risk of HAdV disease.

## 8. Immunotherapy

Based on clinical and immunological observations, the resolution of HAdV viremia and the recovery from HAdV disease is associated with T-cell reconstitution and the appearance of HAdV-specific T cells. Boosting virus-specific T-cell immunity in the first months post-HCT represents an innovative option of intervention [9,41,44]. The adoptive transfer of HAdV immunity is possible by unselected donor lymphocyte infusion (DLI) or by using HAdV-specific T-cells. The efficacy of DLI depends on the variability of the frequency of HAdV-specific T-cells in the donor but has the major drawback of the high risk of severe toxicity due to their content of non-pathogen-specific alloreactive T-cells [45]. To overcome this limit, the current protocols of adoptive immunotherapy are based on the isolation of HAdV-specific T-cells from donor peripheral blood and the expansion of HAdV-specific T-cells ex vivo [46,47,48]. The preparation of HAdV specific-T cells is a complex laboratory process that requires weeks and the availability of a seropositive donor. This may represent a limitation in cases of medical urgent needs or no availability of an authorized laboratory for cellular therapy. This has led to the development of protocols of banked, ready-to-use multiple antiviral T-cell lines, generated by using seropositive third-party donors [49]. In a phase I/II study, 30 patients were treated for HAdV infection or disease with 43 infusions of HAdV-specific CD4+ cells, seven were donor-derived, and 21 third-party derived, obtaining 81% of clinical response and 58% of complete response [50]. A phase II study in 38 patients using third-party-derived pentavalent viruses specific for CMV, EBV, HAdV, HHV-6, and BKV showed an overall efficacy of 92%, with virus-specific cells persisting in circulation for up to 12 weeks after infusion [48]. These recent studies show the potential of immunotherapy in reducing virus-associated morbidity and mortality post-HCT and preventing the toxicity associated with prolonged use of antivirals. Randomized, placebo-controlled studies are ongoing both for prophylaxis and for therapy purposes that could open the way for broader applicability of immunotherapy outside of experimental settings or academic protocols.

## Figures and Tables

**Figure 1 jcm-11-04827-f001:**
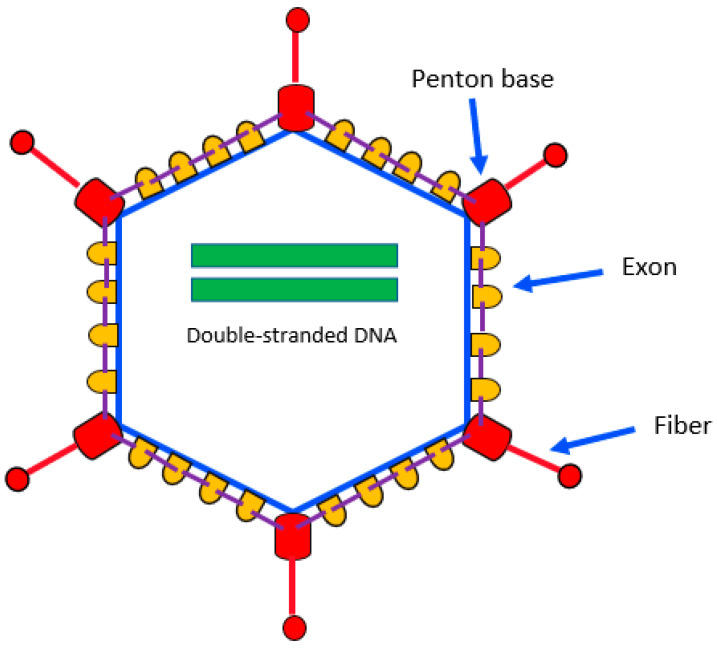
Schematic representation of *Adenovirus* structure.

**Figure 2 jcm-11-04827-f002:**
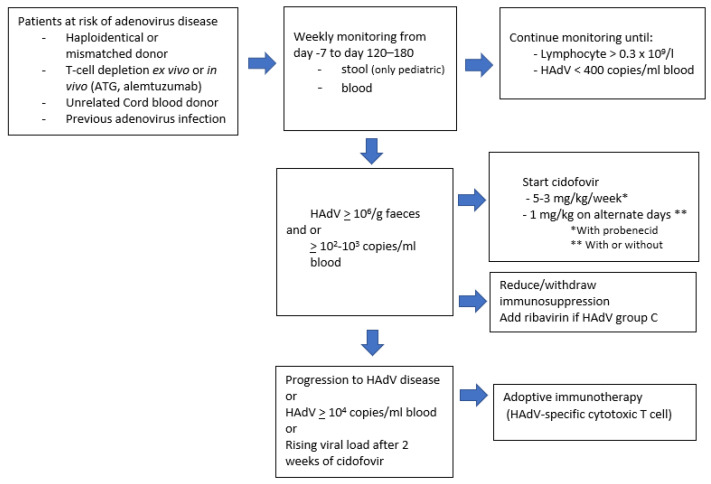
Algorithm for monitoring and treatment of pediatric patients at high risk of adenovirus disease. Legend: ATG, anti-thymocyte globulin; HAdV, human adenovirus.

**Table 1 jcm-11-04827-t001:** List of known human adenovirus species (A–G) and types (1–103). Types 1–51 were identified by serotyping; types (52–103) were identified by genomic sequencing.

Species	Serotypes/Genotypes
A	12, 18, 31, 61
B	3, 7, 11, 14, 16, 21, 34, 35, 50, 55, 66, 68, 76–79
C	1,2,5,6,57,89
D	8–10, 13, 15, 17, 19, 20, 22–30, 32, 33, 36–39,42–49, 51, 53, 54, 56, 58, 59, 60, 63, 64, 65, 67,69–75, 80–88, 90–103
E	4
F	40, 41
G	52

**Table 2 jcm-11-04827-t002:** Risk factors for HAdV infection and HAdV disease in HCT patients.

HAdV Infection	
Children	Allo-HCT with ex vivo or in vivo T-cell depletion Allo-HCT with an unrelated donor Allo-HCT with unrelated cord blood donor Moderate-severe, grade III-IV acute graft versus host disease Lymphopenia ≤ 0.2 × 10^9^/L HAdV ≥ 10^3^ copies/mL stool HAdV shedding in the stool before HCT
Adults	Allo-HCT with unrelated cord blood donor or haploidentical donor Moderate -severe, grade III-IV, acute graft versus host disease Treatment with alemtuzumab
**HAdV disease**	
Children and adults	HAdV viremia ≥ 10^3^ copies/mL blood ≥2 sites/organs HAdV positive

Legend: Allo-HCT, allogeneic hematopoietic cell transplantation; HCT, hematopoietic cell transplantation; HAdV, human adenovirus.

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
