# Peer review of "Adenovirus Infection in Pediatric Hematopoietic Cell Transplantation: A Challenge Still Open for Survival"

_jcm, 2022, doi:10.3390/jcm11164827_

Round 1
Reviewer 1 Report
In this review article, the authors reviewed current research, literature and recommendations on the biology, diagnosis and treatment of adenoviral infections in children undergoing allogeneic hematopoietic cell transplantation. In the paper, the authors also considered risk factors for ADV infections, and presented a simple regimen for patient management depending on the molecular test results obtained. The article is easy to flow, and the language is appropriate.
Comments: Proposal of Fig. 2. modification-I attach the scheme of the proposal.

Author Response
Reviewer 1 Comment: in this review article, the authors reviewed current research, literature and recommendations on the biology, diagnosis and treatment of adenoviral infections in children undergoing allogeneic hematopoietic cell transplantation. In the paper, the authors also considered risk factors for ADV infections, and presented a simple regimen for patient management depending on the molecular test results obtained. The article is easy to flow, and the language is appropriate. Comments: Proposal of Fig. 2. modification-I attach the scheme of the proposal. Reply: We thank the reviewer for the positive opinion expressed on the manuscript. The text has been revised by an english expert in medical publications. Corrections of spelling errors are in red in the text. Figure 2 has been modified on the basis of reviewer suggestions to make the flow chart easier to interpret. We preferred to maintain the vertical disposition instead of horizontal disposition of the figure.
Reviewer 2 Report
In this manuscript entitled “Adenovirus infection in pediatric hematopoietic cell transplantation: a challenge still open for survival”, Cesaro and Porta summarized the most recent studies on adenovirus infection in pediatric patients after hematopoietic cell transplantation. The authors described the biology and subtypes of adenovirus and discussed the risk factors of adenovirus infections and related diseases in transplantation cases. The authors also discussed the recent categories of diagnosis and protocols for prevention and treatment of adenovirus infections and diseases. Most recent publications are included in the discussion and the review discussion is comprehensive.. The manuscript is well-written and the topic is clinically significant. It will have relatively broad audiences. Few typos need to be corrected.
Author Response
Reviewer 2
Comments: In this manuscript entitled “Adenovirus infection in pediatric hematopoietic cell transplantation: a challenge still open for survival”, Cesaro and Porta summarized the most recent studies on adenovirus infection in pediatric patients after hematopoietic cell transplantation. The authors described the biology and subtypes of adenovirus and discussed the risk factors of adenovirus infections and related diseases in transplantation cases. The authors also discussed the recent categories of diagnosis and protocols for prevention and treatment of adenovirus infections and diseases. Most recent publications are included in the discussion and the review discussion is comprehensive. The manuscript is well-written and the topic is clinically significant. It will have relatively broad audiences. Few typos need to be corrected.
Reply:
The authors thank the reviewer for the positive opinion on the manuscript.
The text has been revised by an English expert in medical publications and the spelling errors are in red in the text.
